# Seroprevalence of anti-SARS-CoV-2 antibodies in Japanese COVID-19 patients

**Makoto Hiki[1,2], Yoko Tabe[3], Tomohiko Ai[3], Yuya Matsue[2], Norihiro Harada**📵[4],
**Kiichi Sugimoto[5], Yasushi Matsushita**📵[6], **Masakazu Matsushita[6], Mitsuru Wakita[7],**
**Shigeki Misawa[7], Mayumi Idei[3,8], Takashi Miida[3], Naoto Tamura[6], Kazuhisa Takahashi[4],**
**Toshio Naito**📵[9]*

1 Department of Emergency Medicine, Juntendo University Faculty of Medicine, Tokyo, Japan,
2 Department of Cardiovascular Biology and Medicine, Juntendo University Faculty of Medicine, Tokyo,
Japan, 3 Department of Clinical Laboratory Medicine, Juntendo University Faculty of Medicine, Tokyo,
Japan, 4 Department of Respiratory Medicine, Juntendo University Faculty of Medicine, Tokyo, Japan,
5 Department of Coloproctological Surgery, Juntendo University Faculty of Medicine, Tokyo, Japan,
6 Department of Internal Medicine and Rheumatology, Juntendo University Faculty of Medicine, Tokyo,
Japan, 7 Department of Clinical Laboratory, Juntendo University Hospital, Juntendo University Faculty of
Medicine, Tokyo, Japan, 8 Medical Technology Innovation Center, Juntendo University Faculty of Medicine,
Tokyo, Japan, 9 Department of General Medicine, Juntendo University Faculty of Medicine, Tokyo, Japan

* naito@juntendo.ac.jp

## Abstract

### Objectives

To determine the *seroprevalence* of anti-SARS-CoV-2 *IgG* and IgM antibodies in symptomatic Japanese COVID-19 patients.

### Methods

Serum samples (n = 114) from 34 COVID-19 patients with mild to critical clinical manifestations were examined. The presence and titers of IgG antibody for severe acute respiratory syndrome coronavirus 2 (SARS-CoV-2) were determined by a chemiluminescent microparticle immunoassay (CMIA) using Alinity i SARS-CoV-2 IgG and by an immunochromatographic (IC) IgM/IgG antibody assay using the Anti-SARS-CoV-2 Rapid Test.

### Results

IgG was detected by the CMIA in 40%, 88%, and 100% of samples collected within 1 week, 1–2 weeks, and 2 weeks after symptom onset in severe and critical cases, and 0%, 38%, and 100% in mild/moderate cases, respectively. In severe and critical cases, the positive IgG detection rate with the IC assay was 60% within one week and 63% between one and two weeks. In mild/moderate cases, the positive IgG rate was 17% within one week and 63% between one and two weeks; IgM was positive in 80% and 75% of severe and critical cases, and 42% and 88% of mild/moderate cases, respectively. On the CMIA, no anti-SARS-CoV-2 IgG antibodies were detected in COVID-19 outpatients with mild symptoms within 10 days from onset, whereas 50% of samples from severe inpatients were IgG-

**Data Availability Statement:** The data analyzed in
this research contain sensitive identifying
information (confidential clinical information) and
cannot be shared publicly due to ethical restrictions
imposed by Dr. Atsushi Okuzawa, the Chair of

Juntendo University Hospital Institutional Review Board. Data access requests can be sent to: Dr. Shigeki Aoki, the Chair of Ethical Committee at Juntendo University: E-mail:saoki@juntendo.ac.jp.

**Funding:** The Japanese rock band named GLAY donated funds for this research. The funder had no role in study design, data collection and analysis, decision to publish, or preparation of the manuscript.

**Competing interests:** The authors have declared that no competing interests exist.

positive in the same period. The IC assay detected higher IgM positivity earlier from symptom onset in severe and critical cases than in mild/moderate cases.

## Conclusions

A serologic anti-SARS-CoV-2 antibody analysis can complement PCR for diagnosing COVID-19 14 days after symptom onset.

## Introduction

COVID-19 caused by SARS-CoV-2 infection was first reported in December 2019 in Wuhan, China [1, 2]. It then spread rapidly all over the world, and the World Health Organization (WHO) declared it a pandemic in March 2020. Based on the hundreds of clinical studies reported, approximately 80% of cases show mild symptoms, and approximately 5% of cases, mainly elderly patients and those who have co-existing conditions, develop severe symptoms such as severe respiratory distress syndrome and thromboembolism [3, 4]. Since COVID-19 symptoms are not specific, except for olfactory or taste dysfunction (OTD) [5, 6], diagnoses initially depended solely on PCR tests to detect RNAs of SARS-CoV-2 [7]. However, the sensitivity and specificity were not satisfactory because of sampling issues and the rapid genetic change of the virus [8, 9]. As reported in the previous SARS pandemic cases, samples need to be collected from the lower airway tract for accurate diagnosis. In addition, for the initial PCR tests, the sequences of PCR amplicons were not unique, because the target sequences were the same as those of SARS, MERS, and other types of coronaviruses [10]. Several initial studies reported that the sensitivity of PCR tests was at best 60% [11, 12]. This is problematic for individuals at high risk, such as elderly and immunocompromised patients, since many asymptomatic patients with negative PCR tests can transmit the pathogens without awareness.

As complementary tests to overcome the weakness of PCR tests, various serological tests have been developed. Virus detection by PCR and past exposure detection by SARS-CoV-2-specific antibodies are not competing alternatives, because they should be performed at different time points within their relevant diagnostic windows of clinical development. It has been reported that the incubation time of SARS-CoV-2 might be five days, and IgM antibodies start to be detectable in 5–10 days, and then IgG antibodies start to be detectable within approximately 10 days after symptom onset, with higher titers in severe cases than in mild cases [13, 14]. However, the actual time course of antibody titers has not yet been fully understood. Furthermore, vaccines and detection of neutralizing antibodies are urgently needed for the SARS-CoV-2 pandemic. Although SARS-CoV-2 infection requires the receptor-binding domain (RBD) of the spike protein, little is known about immunoglobulin (Ig) isotypes capable of blocking infection. It has been shown that approximately 86% of S-RBD-binding antibody-positive and 74% of N-protein-binding antibody-positive individuals showed neutralizing capacity during the pandemic in 2020, indicating that detection of N-protein antibodies does not always correlate with the presence of S-RBD-neutralizing antibodies, and thus one should be wary of the extensive use of N-protein-based serological testing for determination of potential COVID-19 immunity [15]. A screening study of human monoclonal antibodies (mAbs) targeting the S-RBD showed that the CT-P59 mAb potently neutralizes SARS-CoV-2 isolates, including the D614G variant, without an antibody-dependent enhancement effect [16]. Another study demonstrated that spike- and RBD-specific IgM, IgG1, and

IgA1 were detected in nearly all patients' plasma/sera at variable levels after infection, and that IgM and IgG1 contributed most to neutralization [17].

In general, RNA viruses including SARS-CoV-2 undergo rapid mutation, resulting in alterations of pathogenicity. Very recently, a SARS-CoV-2 variant with a specific deletion that spanned ORF7b and ORF8 genes was shown to be associated with milder symptoms than those caused by the wild-type virus in Singapore [18]. As such, viral type differs in different regions at different times. For example, at least 116 mutations, including three common mutations, have been identified [19], and the timing of the antibody titers might differ by viral type. Taking these facts into account, the aim of this study was to investigate the time course of the antibody titers in 34 Japanese COVID-19 patients using two currently available serological tests, the chemiluminescent microparticle immunoassay (CMIA) based on SARS-CoV-2 IgG testing (Abbott, Abbott Park, IL, USA) and the immunochromatographic (IC) IgM/IgG antibody assay using the Anti-SARS-CoV-2 Rapid Test (AutoBio, Zhengzhou, China).

## Materials and methods

### Patient cohorts

Laboratory-confirmed COVID-19 cases, including 16 outpatients and 18 hospitalized patients, who were referred to the Juntendo University Hospital, Tokyo, Japan, from March 2 to May 31, 2020 were enrolled in this study. The diagnosis of COVID-19 had been made by prior or same-day PCR-based nasopharyngeal swab testing on the LightMix Modular SARS-CoV (COVID19) N-gene and E-gene assay (Roche Diagnostics, Tokyo, Japan) or the 2019 Novel Coronavirus Detection Kit (Shimadzu, Kyoto, Japan).

As SARS-CoV-2-negative samples, samples obtained at Juntendo University Hospital between November 2017 and December 2018 for routine blood examinations were used.

Clinical data for the patients included major comorbidities, patient-reported symptom onset date, clinical symptoms, and indicators of COVID-19 severity, such as admission to the intensive care unit and requirement for mechanical ventilation. This research complied with all relevant national regulations and institutional policies, was conducted in accordance with the tenets of the Helsinki Declaration, and was approved by the institutional review board (IRB) at Juntendo University Hospital (IRB #20–036). Written, informed consent was obtained from all individuals included in this study.

### Serologic testing for SARS-CoV-2

The CMIA-based SARS-CoV-2 IgG test (cat. # 06R90, Abbott) and the IC IgM/IgG antibody assay using the Anti-SARS-CoV-2 Rapid Test (cat. # RTA0203, AutoBio) were performed using serum samples according to the manufacturers' instructions. The SARS-CoV-2 IgG testing was performed on the Abbott Alinity i in accordance with the manufacturer's specifications. The test is a CMIA for qualitative detection of IgG antibodies against SARS-CoV-2 nucleocapsid protein (NCP) in human serum and plasma. The strength of the response in relative light units reflects the quantity of IgG present, and it is compared to a calibrator to determine the calculated index (specimen/calibrator [S/C]) for a sample (positive at 1.4 or greater).

The anti-SARS-CoV-2 Rapid Test, a rapid lateral flow IC IgM/IgG antibody assay, targets the spike proteins. The presence of only the control line indicates a negative result; the presence of both the control line and the IgM or IgG antibody line indicates a positive result for IgM or IgG antibody, respectively.

Patients were assumed to be seronegative on each day preceding the most recent negative IgG result and to be seropositive on each day following an initial positive result.

## Statistical analysis

Statistical analyses were performed using Stat Flex for Windows (ver. 6.0, Artech, Osaka, Japan). Comparison of the IgG index between the mild to moderate cases and the severe to critical cases was performed using the Mann-Whitney U test. A 2-tailed p value of 0.05 was considered significant.

# Results

## Patients' background characteristics

All 34 patients had a positive RT-PCR SARS-CoV-2 test using a nasopharyngeal swab specimen. The 34 patients were divided into four groups by disease severity based on the WHO criteria [20]: 23 mild, 3 moderate, 4 severe, and 4 critical patients. **Table 1** summarizes the patients' background characteristics. The patients' mean age was less than 50 years in the mild group, whereas it was over 60 years in the other groups. Regarding the male to female ratio, 68% of the patients were male (75% for the severe + critical cases), which is concordant with the previous case series study in the same area, reporting that 76% of the patients were male (96% for the severe cases) [21].

## Sensitivity and specificity of the CMIA SARS-CoV-2 IgG assay and the IC IgM/IgG assay

The CMIA SARS-CoV-2 IgG test using Abbott Alinity i was first examined in a cohort of 16 outpatients (mild) and 18 hospitalized patients (mild to critical). A total of 114 serum samples

**Table 1. Clinical characteristics of patients with COVID-19.**

| Disease severity[a] | | Mild + Moderate | | Severe + Critical | |
|---|---|---|---|---|---|
| | | **Mild** | **Moderate** | **Severe** | **Critical** |
| Patient number (n = 34)[b] | | 68% (23/34) | 9% (3/34) | 12% (4/34) | 12% (4/34) |
| Male, % | | 65% (15/23) | 66% (2/3) | 100% (4/4) | 50% (2/4) |
| Age, y (average) | | 23-82(45.6) | 39-76(61.6) | 56-76(68.3) | 66-78(73.3) |
| Past medical history | | | | | |
| | Hypertension | 13% (3/23) | 0% (0/3) | 0% (0/4) | 75% (3/4) |
| | Hyperlipidemia | 9% (2/23) | 33% (1/3) | 0% (0/4) | 50% (2/4) |
| | Diabetes | 0% (0/23) | 33% (1/3) | 25% (1/4) | 25% (1/4) |
| | Cancer | 0% (0/23) | 0% (0/3) | 0% (0/4) | 75% (3/4) |
| | Renal failure | 0% (0/23) | 0% (0/3) | 0% (0/4) | 25% (1/4) |
| | No known | 78% (18/23) | 33% (1/3) | 75% (3/4) | 0% (0/4) |
| Sample number (n = 114) | | 39% (44/114) | 15% (17/114) | 27% (31/114) | 19% (22/114) |
| Days from onset | | | | | |
| | 0–6 (n = 17) | 12 (12)[c] | 0 | 4 | 1 |
| | 7–13 (n = 17) | 7 (4) | 1 | 6 | 2 |
| | 14–20 (n = 21) | 9 (0) | 4 | 6 | 4 |
| | 21–27 (n = 15) | 4 (0) | 4 | 5 | 2 |
| | 28–34 (n = 18) | 5 (0) | 3 | 4 | 6 |
| | 34–41 (n = 15) | 5 (0) | 3 | 5 | 2 |
| | 42–52 (n = 10) | 2 (0) | 2 | 1 | 5 |

[a] WHO criteria.

[b] 18 inpatients, 16 outpatients.

[c] () outpatients.

**Table 2. Timeline and prevalence of positive CMIA tests using the Alinity i SARS-CoV-2 IgG kit.**

|  |  | Mild + Moderate | | | Severe + Critical | | |
|---|---|---|---|---|---|---|---|
| Patient number | | **26** | | | **8** | | |
| Sample number | | **Total** | **Positive** | **%** | **Total** | **Positive** | **%** |
| Days from onset | | | | | | | |
| | 0–6 | 12 | 0 | 0.0% | 5 | 2 | 40.0% |
| | 7–13 | 8 | 3 | 38.0% | 8 | 7 | 87.5% |
| | 14–20 | 13 | 13 | 100.0% | 10 | 10 | 100.0% |
| | 21–27 | 8 | 8 | 100.0% | 7 | 7 | 100.0% |
| | 28–34 | 8 | 8 | 100.0% | 10 | 10 | 100.0% |
| | 34–41 | 8 | 8 | 100.0% | 7 | 7 | 100.0% |
| | 42–52 | 4 | 4 | 100.0% | 6 | 6 | 100.0% |
| | All | 61 | 44 | 72.1% | 53 | 49 | 92.5% |

were collected. The serum samples were divided into five groups according to sample collection times: within 1 week (n = 17), 1–2 weeks (n = 16), 2–3 weeks (n = 23), 3–4 weeks (n = 15), and >4 weeks after onset (n = 43) (**Table 1**).

**Table 2** shows the change in IgG-positive rates over time in two groups of patients (mild-moderate vs. severe-critical).

All patients became positive 14–20 days after symptom onset. In severe and critical cases, IgG antibody was detected in 40.0% of samples collected within 1 week and 87.5% collected within 1–2 weeks after onset. In contrast, in mild and moderate cases, IgG antibody was not detected within 1 week of onset, and it was detected in 38.0% of samples collected within 1–2 weeks after onset. Of note, all 16 serum samples collected on day 2 –day 9 from outpatients categorized as having mild illness were negative for IgG by CMIA SARS-CoV-2 IgG testing (**Table 3**). To evaluate assay specificity, serum samples obtained before the COVID-19 pandemic era (pre-COVID-19) were tested for IgG seroreactivity (n = 100). All samples were negative, yielding 100% specificity for the SARS-CoV-2 IgG assay (100% in the pre-COVID period).

There is pre-existing SARS-CoV-2 nucleocapsid-specific immunity in people infected by seasonal coronavirus, including HCoV-NL63, -HKU1, and -229E, which have nucleocapsid gene and amino acid (aa) sequence homology with SARS-CoV-2 (20.0%, 27.6%, and 19.1%, respectively) [22]. However, the evaluation study of Abbott SARS CoV-2 IgG has shown no cross-reactivity in samples infected with NL63 (n = 11), HKU1 (n = 7), and 229E (n = 3) [23]. Public Health England has also reported 100% specificity for Abbott SARS-CoV-2 IgG in the seasonal coronavirus positive samples (n = 11) [24].

IgM and IgG antibodies for SARS-CoV-2 detected by an IC assay using the Anti-SARS-CoV-2 Rapid Test were next examined. All patients (100%) were seropositive for IgG and/or IgM within 14 days after symptom onset (**Table 4**). In severe and critical cases, IgM antibody was detected in 80.0% (4/5) of samples collected within 1 week of onset, 75.0% (6/8) in 1–2

**Table 3. Timeline and prevalence of positive CMIA tests using the Alinity i SARS-CoV-2 IgG kit in outpatients.**

| Sample number (n = 16) | | Total | Positive | % |
|---|---|---|---|---|
| Days from onset | | | | |
| | 0–6 | 12 | 0 | 0.0% |
| | 7–13 | 4 | 0 | 0.0% |
| | All | 16 | 0 | 0.0% |

**Table 4. Timeline and prevalence of positive IC tests using the Anti-SARS-CoV-2 Rapid Test.**

| | | IgG | | | | | | IgM | | | | | | IgG+IgM | | | | | |
|---|---|---|---|---|---|---|---|---|---|---|---|---|---|---|---|---|---|---|---|
| | | Mild + Moderate | | | Severe + Critical | | | Mild + Moderate | | | Severe + Critical | | | Mild + Moderate | | | Severe + Critical | | |
| Patient number | | 26 | | | 8 | | | 26 | | | 8 | | | 26 | | | 8 | | |
| Sample number | Total | Positive | % | Total | Positive | % | Total | Positive | % | Total | Positive | % | Total | Positive | % | Total | Positive | % |
| Days from onset | | | | | | | | | | | | | | | | | | |
| 0–6 | 12 | 2 | 16.7% | 5 | 3 | 60.0% | 12 | 5 | 41.7% | 5 | 4 | 80.0% | 12 | 5 | 41.7% | 5 | 4 | 80.0% |
| 7–13 | 8 | 5 | 62.5% | 8 | 5 | 62.5% | 8 | 7 | 87.5% | 8 | 6 | 75.0% | 8 | 7 | 87.5% | 8 | 7 | 87.5% |
| 14–20 | 13 | 13 | 100.0% | 10 | 10 | 100.0% | 13 | 12 | 92.3% | 10 | 9 | 90.0% | 13 | 13 | 100.0% | 10 | 10 | 100.0% |
| 21–27 | 8 | 8 | 100.0% | 7 | 7 | 100.0% | 8 | 8 | 100.0% | 7 | 7 | 100.0% | 8 | 8 | 100.0% | 7 | 7 | 100.0% |
| 28–34 | 8 | 8 | 100.0% | 10 | 10 | 100.0% | 8 | 5 | 62.5% | 10 | 10 | 100.0% | 8 | 8 | 100.0% | 10 | 10 | 100.0% |
| 34–41 | 8 | 8 | 100.0% | 7 | 7 | 100.0% | 8 | 6 | 75.0% | 7 | 7 | 100.0% | 8 | 8 | 100.0% | 7 | 7 | 100.0% |
| 42–52 | 4 | 4 | 100.0% | 6 | 6 | 100.0% | 4 | 3 | 75.0% | 6 | 6 | 100.0% | 4 | 4 | 100.0% | 6 | 6 | 100.0% |
| All | 61 | 48 | 78.7% | 53 | 48 | 90.6% | 61 | 46 | 75.4% | 53 | 49 | 92.5% | 61 | 53 | 86.9% | 53 | 51 | 96.2% |

Regarding the disagreement between the CMIA SARS-CoV-2 IgG assay and the IC Anti-SARS-CoV-2 Rapid Test, 0% (0/114) were CMIA assay-positive and IC test-negative, and 10% (11/114) were CMIA-negative and IC-positive.

weeks, 90.0% (9/10) in 2–3 weeks, and 100% (30/30) collected >3 weeks after onset. The detection rates for IgG antibody were 60.0% (3/5) less than 1 week after onset, 62.5% (5/8) in 1–2 weeks, and 100% in the second week and thereafter. In mild and moderate cases, IgM antibody was detected in 41.7% (5/12) of samples collected within 1 week of onset, 87.5% (7/8) in 1–2 weeks, 92.3% (12/13) in 2–3 weeks, and 100% (8/8) in 3–4 weeks. Of note, the IgM-positive rates decreased 4 weeks after onset; 62.5% (5/8) in 4–5 weeks, 75.0% (6/8) in 5–6 weeks, and 75.0% (3/4) >6 weeks after onset. The corresponding detection rates for IgG antibody were 16.7% (2/12) within 1 week, 62.5% (5/8) in 1–2 weeks, and 100% (41/41) in >2 weeks after onset. Eighty-two percent (98/114) of IgG antibody-positive samples were also positive for IgM antibody with the IC assay. Six of eight IgG-positive and IgM-negative samples were obtained from mild to moderate cases more than 4 weeks after onset. In addition, there were 6 IgM-positive and IgG-negative samples, 4 of which were taken within 1 week after onset, and 2 were taken within 1–2 weeks of symptom onset. In the IC assays in the 100 non-COVID-19 serum samples, all samples were negative for IgM and IgG. Thus, the specificity of the IC assay was 100%.

### Longitudinal dynamics of IgG titers

Using the calculated index of the CMIA SARS-CoV-2 IgG assay, the longitudinal dynamics of the serum IgG (114 samples) levels of 34 patients who were positive for SARS-CoV-2 by PCR were analyzed. As shown in **Fig 1**, IgG antibody levels increased every week after symptom onset. The IgG antibody levels were significantly higher in severe to critical cases than in mild to moderate cases until 7 days.

The chronological changes of the IgG index in eight inpatients were plotted against the time of the tests (Fig 2). Table 5 summarizes the patients' clinical background characteristics. There were no significant differences in IgG index levels between the surviving and deceased cases.

### Discussion

Rapid PCR tests and serological assays are two important tools in managing the COVID-19 pandemic. Highly sensitive serology assays are required to complement PCR methods whose

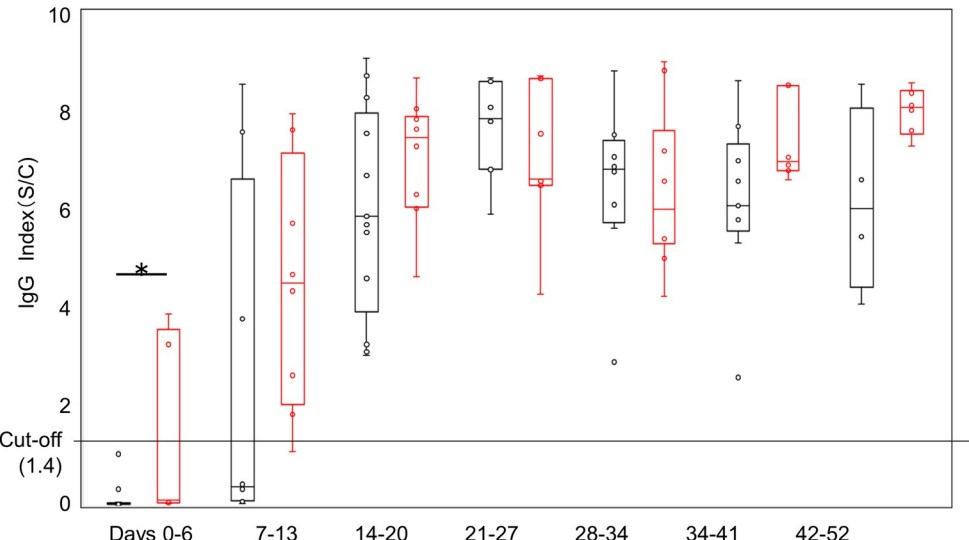

**Fig 1. Seroprevalence of antibodies to SARS-CoV-2.** IgG index of the CMIA Alinity i SARS-CoV-2 assay for SARS-CoV-2 PCR-positive patient samples for the indicated weekly timeframes after onset of symptoms. The black line and dots represent mild to moderate cases, and the red line and dots represent severe to critical cases. *$P < 0.05$.

detection rates decrease 2 weeks after symptom onset [25] not to miss a large number of patients with minimal symptoms to prevent virus transmission [26].

This study investigated the time courses of serological tests in 34 Japanese patients using the CMIA SARS-CoV-2 IgG assay (Abbott) and the IC IgM/IgG assay (AutoBio).

Both assays demonstrated perfect specificity, with 92% agreement on IgG positivity. These tests showed 100% positivity for IgG from 2 weeks after symptom onset and remained positive until 2 months in mild to critical cases, which is consistent with the previous reports [27–30].

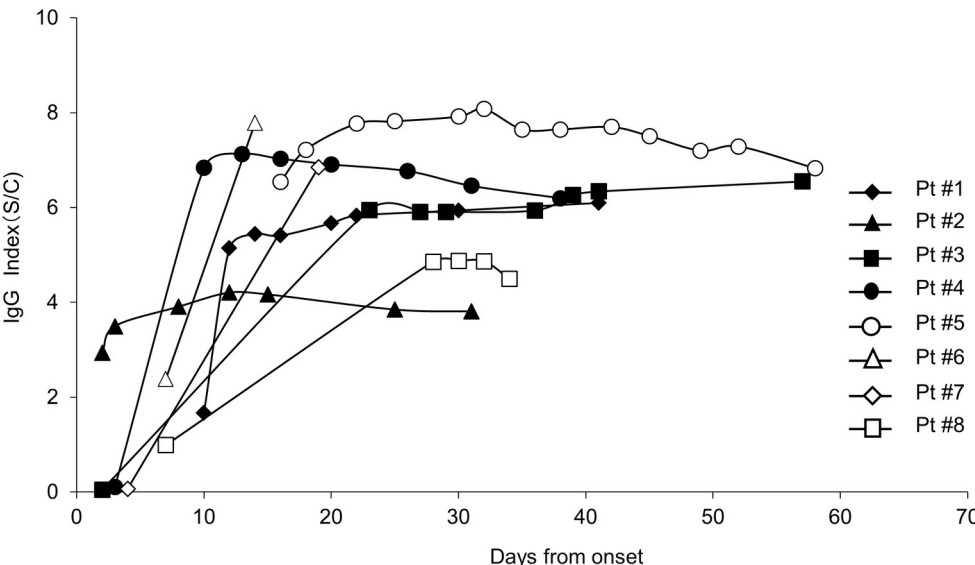

**Fig 2. Longitudinal changes of antibodies against SARS-CoV-2 in severe cases.** The cut-off index was tested using Alinity i SARS-CoV-2in eight severe patients. The IgG index was plotted as a function of days after onset. Closed symbols depict surviving cases, and open symbols depict deceased cases.

**Table 5. Clinical characteristics of patients with severe COVID-19.**

| Patient # | Outcome | Age (y) | Sex | Past medical history |
|---|---|---|---|---|
| 1 | Cure / discharge | 57 | M | None |
| 2 | Cure / discharge | 77 | M | Diabetes, Rheumatoid arthritis, Pneumonia |
| 3 | Cure / discharge | 75 | M | Prostatic hypertrophy |
| 4 | Cure / discharge | 66 | M | None |
| 5 | Dead | 76 | M | Hypertension, Diabetes, Cancer |
| 6 | Dead | 78 | F | None |
| 7 | Dead | 64 | F | Hyperlipidemia, Cancer |
| 8 | Dead | 67 | M | Hypertension, Renal failure |

All COVID-19 outpatients with mild symptoms were seronegative within 10 days after onset. In contrast, 40% and 87.5% of the severe to critical patients became positive at 0–6 days and 7–13 days after onset, respectively. These results do not agree with the previous results demonstrating that IgG levels tested by the same CMIA SARS-CoV-2 IgG assay did not differ with patients' disease severity [31]. The previous studies using the CMIA SARS-CoV-2 IgG assay from the USA reported that 53.1% and 82.4% of the COVID-19 patients were IgG-positive in 7 days and 10 days after onset, respectively [32], or 30.0% positive at 3–7 days and 47.8% positive at 8–13 days after onset [33]. These studies did not analyze the results according to severity. In mild cases, the infection is limited to the nasopharyngeal cells. Thus, RT-PCR tests can detect the pathogen's RNA before immune systems produce antibodies. This delay is well known and described in the Interim Guidelines for COVID-19 Antibody Testing [34].

The duration of SARS-CoV-2 RNA shedding is known to depend on the patient's medical condition. Previous studies of COVID-19 patients reported that the median duration of viral shedding was 20 days in survivors (range, 8–37 days) [35]. In contrast, the median duration of viral shedding in mild cases was 14 days [34]. The IgG levels in a high proportion of individuals who recovered from SARS-CoV-2 infection have been reported to start decreasing within 2–3 months after infection [36, 37].

The studies from China, Europe, and the USA reported that males aged 50 years or older are at increased risk of severe COVID-19 [38], which is concordant with the present findings.

In the present study, IgG titers remained at significantly elevated levels for 2 months, regardless of disease severity. These results indicate that IgG serologic tests could be used as a complementary test to PCR to diagnose COVID-19 from 14 days after symptom onset, including patients with mild symptoms without dyspnea, remaining antibody-positive for at least 2 months after infection. On the other hand, the IgM positivity on the IC assay of the Anti-SARS-CoV-2 Rapid Test decreased at 4 weeks after symptom onset in the mild cases.

The CMIA SARS-CoV-2 IgG assay, detecting anti-nucleocapsid antibody, has been reported to produce false-positive reactions [39], which might be either because they constitute false-positive reactions [40] or because of the lack of a detectable humoral response to S1 that can occur in asymptomatic or mild infections [41, 42]. However, in the present study, all of the disagreement between the CMIA SARS-CoV-2 IgG assay and the IC Anti-SARS-CoV-2 Rapid Test, detecting anti-S-RBD antibody, were CMIA-negative and IC-positive cases, which indicates no false-positive reactions of CMIA-positive and IC-negative cases, at least in the present study.

Concerning the effector mechanisms, it has been reported that antibodies of different subclasses activate different effector mechanisms in response to RBD antigens, and detection of anti-RBD subclasses might be important in clinical research [43]. Although a combination of IgG1 and IgG2 subclasses against the RBD may enhance this effect and reduce disease severity

[43], and an antibody cocktail against the pseudogene has been shown in in vitro experiments [44], the role of anti-RBD antibodies in the immune response against SARS-CoV-2 is still controversial [45].

Several mutations of SARS-CoV-2 have been reported, including the ones observed in East Asia, which are associated with milder symptoms than those of the wild-type virus infection [18]. Indeed, the incidence and mortality of COVID-19 varied in different countries/territories [46].

The major limitations of this study were the small number of patients in a single hospital and the short study period. Additional multicenter, longitudinal, serological studies with more detailed profiling by medical condition, including patients with mild and severe symptoms and asymptomatic patients, in different endemic areas are urgently needed to determine the duration of antibody-mediated immunity.

The present findings underscore the importance of highly sensitive serological tests to detect SARS-CoV-2 in mildly symptomatic or asymptomatic populations early in the infection. Several simultaneous or repeated tests may be able to overcome an individual test's limited sensitivity, and validation of further strategies is needed.

## Acknowledgments

The authors would like to thank Kaori Saito, Masayoshi Chonan, Koji Tsuchiya, Takaaki Kawakami, Suzuka Ishikawa, and Natsumi Itakura for technical support. The authors would also like to thank Tomohiro Akazawa for useful discussion. Finally, the authors are grateful to Sakiko Miyazaki, Tsuyoshi Ueno, and Juntendo COVID team members for data collection.

## Author Contributions

**Conceptualization:** Yoko Tabe, Norihiro Harada, Kazuhisa Takahashi.

**Data curation:** Makoto Hiki, Yuya Matsue, Kiichi Sugimoto, Yasushi Matsushita, Masakazu Matsushita, Mitsuru Wakita, Shigeki Misawa, Mayumi Idei, Takashi Miida.

**Methodology:** Mitsuru Wakita, Shigeki Misawa, Mayumi Idei.

**Resources:** Naoto Tamura.

**Supervision:** Yoko Tabe, Takashi Miida, Kazuhisa Takahashi.

**Writing – original draft:** Makoto Hiki, Tomohiko Ai.

**Writing – review & editing:** Yoko Tabe, Toshio Naito.

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
