## [Decision Letter · Decision Letter 0]

6 Jan 2021

PONE-D-20-31485

Seroprevalence of anti-SARS-CoV-2 antibodies in Japanese COVID-19 patients

PLOS ONE

Dear Dr. Naito,

Thank you for submitting your manuscript to PLOS ONE. After careful consideration, we feel that it has merit but does not fully meet PLOS ONE’s publication criteria as it currently stands. Therefore, we invite you to submit a revised version of the manuscript that addresses the points raised during the review process.

The anonymised comments to author from all reviewers, are enclosed below, asked for substantial revisions in the next version. 

We look forward to receiving your revised manuscript.

Kind regards,

Chandrabose Selvaraj, Ph.D.

Academic Editor

PLOS ONE

Journal Requirements:

2. Thank you for stating in the text of your manuscript that your study "was approved by the institutional review board (IRB) at Juntendo University Hospital (IRB #20-036). Written, informed consent was obtained from all individuals included in this study." Please also add this information to your ethics statement in the online submission form.

3. PLOS ONE requires experimental methods to be described in enough detail to allow suitably skilled investigators to fully replicate and evaluate your study. See https://journals.plos.org/plosone/s/submission-guidelines#loc-materials-and-methods for more information.

To comply with PLOS ONE submission guidelines, in your Methods section, please provide the catalog numbers of the Serologic tests for SARS-CoV-2, in the methods section of your manuscript.

4.PLOS requires an ORCID iD for the corresponding author in Editorial Manager on papers submitted after December 6th, 2016. Please ensure that you have an ORCID iD and that it is validated in Editorial Manager. To do this, go to ‘Update my Information’ (in the upper left-hand corner of the main menu), and click on the Fetch/Validate link next to the ORCID field. This will take you to the ORCID site and allow you to create a new iD or authenticate a pre-existing iD in Editorial Manager. Please see the following video for instructions on linking an ORCID iD to your Editorial Manager account: https://www.youtube.com/watch?v=_xcclfuvtxQ

<h1>** **</h1>

Reviewers' comments:

Reviewer's Responses to Questions

**Comments to the Author**

1. Is the manuscript technically sound, and do the data support the conclusions?

Reviewer #1: Partly

Reviewer #2: Partly

Reviewer #3: Yes

2. Has the statistical analysis been performed appropriately and rigorously? 

Reviewer #1: Yes

Reviewer #2: Yes

Reviewer #3: Yes

3. Have the authors made all data underlying the findings in their manuscript fully available?

Reviewer #1: Yes

Reviewer #2: Yes

Reviewer #3: Yes

4. Is the manuscript presented in an intelligible fashion and written in standard English?

Reviewer #1: Yes

Reviewer #2: Yes

Reviewer #3: Yes

5. Review Comments to the Author

Reviewer #1: The authors have provided the seroprevalence of anti-SARS-CoV-2 IgG and IgM antibodies in 34 symptomatic Japanese COVID-19 patients and concluded that a serologic anti-SARS-CoV-2 antibody analysis can complement PCR for diagnosing COVID-19 14 days after symptom onset.

The immunological characteristic of COVID-19 patients has been intensively studied, such as cross-reactivity of anti-RBD, anti-Spike and anti-nucleocapsid and effector mechanisms of different subclasses of antibody. However, the author failed to summarize all the knowledge in the introduction.

There is pre-existing SARS-CoV-2 nucleocapsid-specific immunity in people infected by seasonal “common cold” HCoV-OC43 and HCoV-NL63 viruses. Therefore, the anti-SARS-CoV-2 antibody test requires negative control samples including seasonal "common cold" HCoV-OC43 and HCoV-NL63 viruses. Furthermore, effector mechanisms should be discussed. Antibodies of different subclasses activate different effector mechanisms in response to RBD antigens. Therefore, a combination of IgG1 and IgG2 subclasses against the RBD may enhance this effect and reduce disease severity. The detection of anti-RBD subclasses is important to clinical research. [Tackling COVID19 by exploiting pre-existing cross-reacting spike-specific immunity. Mol Ther. 2020 Nov 4; 28(11): 2314–2315.]

The authors used Abbott SARS-CoV-2 IgG assay which detects anti-nucleocapsid antibody. The Abbott assay results constitute false-positive reactions due to cross-reactivity to nucleocapsid as reported in Lancet [Testing for responses to the wrong SARS-CoV-2 antigen?. Lancet 2020; published online August 28. http://dx.doi.org/10.1016/S0140-6736(20)31830-4.] and Science Translational Medicine [https://stm.sciencemag.org/content/12/559/eabc3103/tab-e-letters]. To further clarify this issue, the study should include anti-SARS-CoV-2 RBD antibody detection methods.

Reviewer #2: 1. Authors have mentioned that IgG antibodies were detectable in 5- 10 days, is it possible to detect the antibodies with a high percentage of antibodies in asymptomatic patients?

2. Authors have mentioned that clinical data includes patients that have major comorbidities, what are the major comorbidities were diagnosed in COVI-19 patients, is these comorbidities play a key role in antibody production?

3. Why have authors selected only male candidates for the selection for this study?

4. In table 2, Compared to severe and critical samples shows a higher percentage of positive results than mild and moderate is this kit only gives accuracy in severe/critical cases?

5. Authors have mentioned that these two kits are complementary PCR for COVID-19 detection. But the SARS-CoV-2 infected individuals are identified by RT-PCR and the nasopharyngeal sampling was positive in approximately 89% (95% CI 83 to 93) of tests within 4 days of either symptom onset. But these two kits do not detect the antibodies in the mild cases. Then how authors have justified these kits are used as complementary methods.

6. In table 3, The IC result shows, the percentage of IgG and IgM in mild and moderate shows very less content in the starting period of the study. Is this starting period the initial stage of the COVID-19 infection?

7. Compared to PCR how much accuracy is noticed in these two kits. Any evidence?

8. If the infection was detected in the initial stage it could be more effective, but these two kits do not detect any antibodies in initial or mild to moderate cases, how could it be more effective than PCR?

9. Did authors find differences in the percentage of IgG and IgM in patients with comorbidities and without any comorbidity?

Reviewer #3: Hiki and team had summarized their findings on seroprevalence of anti-SARS-CoV-2 antibodies in Japanese COVID-19 patients using CMIA. According to their report, the antibodies were developed within 2 weeks of onset of the disease; however, there were numbers of patients who were at severe to critical stage after the development of the IgG. It is suggested to report the clinical manifestations of the included patients and correlate their recovery with the clinical improvement and developed antibodies.

6. PLOS authors have the option to publish the peer review history of their article (what does this mean?). If published, this will include your full peer review and any attached files.

Reviewer #1: No

Reviewer #2: No

Reviewer #3: No

---

## [Author Response · Author response to Decision Letter 0]

8 Mar 2021

COMMENTS by Reviewer #1: 

The authors have provided the seroprevalence of anti-SARS-CoV-2 IgG and IgM antibodies in 34 symptomatic Japanese COVID-19 patients and concluded that a serologic anti-SARS-CoV-2 antibody analysis can complement PCR for diagnosing COVID-19 14 days after symptom onset.

The immunological characteristic of COVID-19 patients has been intensively studied, such as cross-reactivity of anti-RBD, anti-Spike and anti-nucleocapsid and effector mechanisms of different subclasses of antibody. However, the author failed to summarize all the knowledge in the introduction.

ANSWERS:

Thank you for your time and comments.

The aim of this study was to report the seroprevalence detected by several commercially available assays, and there is no evidence showing that these assays can detect so-called “neutralizing antibodies.” Therefore, attempting to dissect “the effector mechanisms of different class of antibodies” is beyond the scope of this study.

Although we agree that understanding of neutralizing antibodies is important, the effector mechanism stated by the reviewer is still controversial, as shown in many of the most recent publications (e.g., PMID: 33436577, 32796155, 33442694, 33412089, 33367897).

We added text regarding these topics in the introduction. (L79-91)

COMMENTS (continued):

There is pre-existing SARS-CoV-2 nucleocapsid-specific immunity in people infected by seasonal “common cold” HCoV-OC43 and HCoV-NL63 viruses. Therefore, the anti-SARS-CoV-2 antibody test requires negative control samples including seasonal "common cold" HCoV-OC43 and HCoV-NL63 viruses. 

ANSWER:

Thank you for the important suggestion. We included the cross-reactivity of the N antibody reagent we used. (L175-183)

COMMENTS (continued):

Furthermore, effector mechanisms should be discussed. Antibodies of different subclasses activate different effector mechanisms in response to RBD antigens. Therefore, a combination of IgG1 and IgG2 subclasses against the RBD may enhance this effect and reduce disease severity. The detection of anti-RBD subclasses is important to clinical research. [Tackling COVID19 by exploiting pre-existing cross-reacting spike-specific immunity. Mol Ther. 2020 Nov 4; 28(11): 2314–2315.]

ANSWER:

Although this topic is beyond the scope of this study, and the role of anti-RBD antibodies in the immune response against SARS-CoV-2 is still controversial, we added relevant information in the Discussion section.(L277-282).

COMMENTS (continued):

The authors used Abbott SARS-CoV-2 IgG assay which detects anti-nucleocapsid antibody. The Abbott assay results constitute false-positive reactions due to cross-reactivity to nucleocapsid as reported in Lancet [Testing for responses to the wrong SARS-CoV-2 antigen?. Lancet 2020; published online August 28. http://dx.doi.org/10.1016/S0140-6736(20)31830-4.] and Science Translational Medicine [https://stm.sciencemag.org/content/12/559/eabc3103/tab-e-letters]. To further clarify this issue, the study should include anti-SARS-CoV-2 RBD antibody detection methods.

ANSWER:

Following the reviewer’s suggestion, we discussed the possibility of false-positive results in our study.(L270-276)

Reviewer #2: 

COMMENT 1. Authors have mentioned that IgG antibodies were detectable in 5- 10 days, is it possible to detect the antibodies with a high percentage of antibodies in asymptomatic patients?

ANSWER: This is an important point for epidemiological studies. Unfortunately, our study did not include asymptomatic individuals. A few studies have reported the seroprevalence in asymptomatic individuals or carriers.(PMID: 33411805) However, it is almost impossible to check RT-PCR and subsequent antibody tests since nobody can tell when the individuals are infected.

COMMENT 2. Authors have mentioned that clinical data includes patients that have major comorbidities, what are the major comorbidities were diagnosed in COVI-19 patients, is these comorbidities play a key role in antibody production?

ANSWER: We showed the additional data for the clinical characteristics of severe COVID-19 patients with their longitudinal changes in IgG antibodies against SARS-CoV-2. No specific associations between comorbidities and antibody production were observed.(L222-226, Figure 2, Table 5).

COMMENT 3. Why have authors selected only male candidates for the selection for this study?

ANSWER: According to a case series in the same area (next to our hospital), 75.6% of the patients were male (95.5% for the severe cases)(PMID: 32970757). (L150-153)

COMMENT 4. In table 2, Compared to severe and critical samples shows a higher percentage of positive results than mild and moderate is this kit only gives accuracy in severe/critical cases?

ANSWER: RT-PCR tests can detect the pathogen’s RNA before immune systems produce antibodies. This delay is well known and described in the Interim Guidelines for COVID-19 Antibody Testing (L251-255). Since the infection might be limited to the nasopharyngeal cells in mild cases, the delay was longer in the mild cases than in the severe cases. However, from 2 weeks after symptom onset, these tests showed 100% positivity and remained positive until two months, both in mild to severe/critical cases. This was described in L164-173.

COMMENT 5. Authors have mentioned that these two kits are complementary PCR for COVID-19 detection. But the SARS-CoV-2 infected individuals are identified by RT-PCR and the nasopharyngeal sampling was positive in approximately 89% (95% CI 83 to 93) of tests within 4 days of either symptom onset. But these two kits do not detect the antibodies in the mild cases. Then how authors have justified these kits are used as complementary methods.

ANSWER: There are no tests with 100% sensitivity and 100% specificity. Specifically, SARS-CoV-2 is a new pathogen, and there is no true standard method. Indeed, nobody knows the true sensitivity and specificity of RT-PCR tests, either. Since there are no other definite diagnostic methods, the sensitivity and specificity of all antibody tests are currently determined using RT-PCR tests as the reference. However, there are some reported cases of RT-PCR-negative and antibody-positive (e.g., PMID: 32335175). Thus, antibody tests can be complementary to RT-PCR tests. 

COMMENT 6. In table 3, The IC result shows, the percentage of IgG and IgM in mild and moderate shows very less content in the starting period of the study. Is this starting period the initial stage of the COVID-19 infection?

ANSWER: As the reviewer indicates, the infection might be limited to the nasopharyngeal cells in mild cases. Thus, RT-PCR tests can detect pathogen’s RNA before immune systems produce antibodies. This delay is well known and described in the Interim Guidelines for COVID-19 Antibody Testing (https://www.cdc.gov/coronavirus/2019-ncov/lab/resources/antibody-tests-guidelines.html).(L251-255).

COMMENT 7. Compared to PCR how much accuracy is noticed in these two kits. Any evidence?

ANSWER: Unfortunately, nobody knows the answer, since the sensitivity of antibody tests is determined based on RT-PCR tests. However, the relationship between viral loads determined by RT-PCR tests (e.g., Ct values) and infectivity is not yet understood. Thus, we do not know what is true. 

COMMENT 8. If the infection was detected in the initial stage it could be more effective, but these two kits do not detect any antibodies in initial or mild to moderate cases, how could it be more effective than PCR?

ANSWER: We cannot say that the antibody tests are more effective than PCR in the initial period of COVID-19 infection, especially in mild to moderate cases. Please see the answer for COMMENT 6.

COMMENT 9. Did authors find differences in the percentage of IgG and IgM in patients with comorbidities and without any comorbidity?

ANSWER: We observed almost the same positivity for IgG and IgM. No specific association was observed between the comorbidities in the severe and critical cases (Table 5) and positivity of IgG and IgM.

Reviewer #3: 

COMMENTS: Hiki and team had summarized their findings on seroprevalence of anti-SARS-CoV-2 antibodies in Japanese COVID-19 patients using CMIA. According to their report, the antibodies were developed within 2 weeks of onset of the disease; however, there were numbers of patients who were at severe to critical stage after the development of the IgG. It is suggested to report the clinical manifestations of the included patients and correlate their recovery with the clinical improvement and developed antibodies.

ANSWER: Thank you for the important comments. As suggested, we added the clinical characteristics of the severe and critical patients and the association between clinical improvement and the chronological changes of the IgG index (L222-226, Figure 2, Table 5).

---

## [Decision Letter · Decision Letter 1]

19 Mar 2021

Seroprevalence of anti-SARS-CoV-2 antibodies in Japanese COVID-19 patients

PONE-D-20-31485R1

Dear Dr. Naito,

We’re pleased to inform you that your manuscript has been judged scientifically suitable for publication and will be formally accepted for publication once it meets all outstanding technical requirements.

Kind regards,

Chandrabose Selvaraj, Ph.D.

Academic Editor

PLOS ONE

Additional Editor Comments (optional):

Reviewers' comments:

Reviewer's Responses to Questions

**Comments to the Author**

1. If the authors have adequately addressed your comments raised in a previous round of review and you feel that this manuscript is now acceptable for publication, you may indicate that here to bypass the “Comments to the Author” section, enter your conflict of interest statement in the “Confidential to Editor” section, and submit your "Accept" recommendation.

Reviewer #1: All comments have been addressed

Reviewer #2: All comments have been addressed

Reviewer #3: All comments have been addressed

2. Is the manuscript technically sound, and do the data support the conclusions?

Reviewer #1: Yes

Reviewer #2: Yes

Reviewer #3: Yes

3. Has the statistical analysis been performed appropriately and rigorously? 

Reviewer #1: Yes

Reviewer #2: Yes

Reviewer #3: Yes

4. Have the authors made all data underlying the findings in their manuscript fully available?

Reviewer #1: Yes

Reviewer #2: Yes

Reviewer #3: Yes

5. Is the manuscript presented in an intelligible fashion and written in standard English?

Reviewer #1: Yes

Reviewer #2: Yes

Reviewer #3: Yes

6. Review Comments to the Author

Reviewer #1: The authors have provided the seroprevalence of anti-SARS-CoV-2 IgG and IgM antibodies in 34 symptomatic Japanese COVID-19 patients and concluded that a serologic anti-SARS-CoV-2 antibody analysis can complement PCR for diagnosingCOVID-19 14 days after symptom onset.

The immunological characteristic of COVID-19 patients has been intensively discussed, such as cross-reactivity of anti-RBD, anti-Spike and anti-nucleocapsid and effector mechanisms of different subclasses of antibody.

Reviewer #2: authors have explained all the quires with significant justification to the reviewer comments, and authors have used well standard statistical methods and maintain the good language throughout the manuscript.

Reviewer #3: The authors addressed the comments made to improve the quality of the article to meet the journal standard. Thank you.

7. PLOS authors have the option to publish the peer review history of their article (what does this mean?). If published, this will include your full peer review and any attached files.

Reviewer #1: **Yes: **Yang Xu, MD, PhD

Reviewer #2: No

Reviewer #3: No

---

## [Editor Report · Acceptance letter]

24 Mar 2021

PONE-D-20-31485R1 

Seroprevalence of anti-SARS-CoV-2 antibodies in Japanese COVID-19 patients 

Dear Dr. Naito:

I'm pleased to inform you that your manuscript has been deemed suitable for publication in PLOS ONE. Congratulations! Your manuscript is now with our production department. 

Kind regards, 

on behalf of

Dr. Chandrabose Selvaraj 

Academic Editor

PLOS ONE